# Efficacy and durability of bovine virus diarrhea (BVD) virus killed vaccine adjuvanted with monolaurin

Maha Raafat Abd El Fadeel[1☯], Eman M. Soliman[2☯], Ahmad Mohammad Allam[3☯]*, Mohamed F. ElKersh[4☯¤a], Rehab Mahmoud Abd El-Baky[5☯¤b], Ahmad Mustafa[6☯]

1 Veterinary Serum and Vaccine Research Institute, Abbasia, Cairo, Egypt, 2 Central Laboratory for Evaluation of Veterinary Biologics, Agriculture Research Center (CLEVB/ARC), Abbasia, Cairo, Egypt, 3 Department of Parasitology and Animal Diseases, Veterinary Research Institute, National Research Centre, Dokki, Giza, Egypt, 4 Animal Health Institute AHRI, Agriculture Research Center ARC, Cairo, Egypt, 5 Department of Microbiology and Immunology, Faculty of Pharmacy, Minia University, Minia, Egypt, 6 Faculty of Engineering, October University for Modern Sciences and Arts (MSA), Giza, Egypt

☯ These authors contributed equally to this work.
¤a Current address: Ministry of Agriculture and Land Reclamation, Cairo, Egypt
¤b Current address: Department of Microbiology and Immunology, Faculty of Pharmacy, Deraya University, Minia, Egypt
* ahmdallam@gmail.com

**Data Availability Statement:** All relevant data are within the manuscript and its Supporting Information files.

**Funding:** The author(s) received no specific funding for this work.

## Abstract

The bovine virus diarrhea virus (BVDV) causes reproductive, enteric, and respiratory diseases. Vaccination is essential in increasing herd resistance to BVDV spread. The selection of an adjuvant is an important factor in the success of the vaccination process. Monolaurin or glycerol monolaurate is a safe compound with an immunomodulatory effect. This study aimed to evaluate the efficacy of monolaurin as a novel adjuvant. This was examined through the preparation of an inactivated BVDV (NADL strain) vaccine adjuvanted with different concentrations of monolaurin and compared with the registered available locally prepared polyvalent vaccine (Pneumo-4) containing BVD (NADL strain), BoHV-1 (Abou Hammad strain), BPI3 (strain 45), and BRSV (strain 375L), and adjuvanted with aluminum hydroxide gel. The inactivated BVDV vaccine was prepared using three concentrations, 0.5%, 1%, and 2%, from monolaurin as adjuvants. A potency test was performed on five groups of animals. The first group, which did not receive vaccination, served as a control group while three other groups were vaccinated using the prepared vaccines. The fifth group received the Pneumo-4 vaccine. Vaccination response was monitored by measuring viral neutralizing antibodies using enzyme-linked immunosorbent assay (ELISA). It was found that the BVD inactivated vaccine with 1% and 2% monolaurin elicited higher neutralizing antibodies that have longer-lasting effects (nine months) with no reaction at the injection site in comparison to the commercial vaccine adjuvanted by aluminum hydroxide gel.

## Introduction

The bovine virus diarrhea virus (BVDV), an enveloped, single-strand RNA virus (genus *Pestivirus*, family *Flaviviridae*), can infect a wide range of animals such as cattle, sheep, goats, pigs,

**Competing interests:** The authors have declared that no competing interests exist.

and domestic and wild ruminant species. It causes reproductive, enteric, and respiratory diseases. As BVDV infection is associated with significant immune dysfunction resulting in a variety of infections, it can cause either acute or life-long persistent infection (PI) and a significant economic loss of dairy- and beef-producing animals [1, 2]. Regarding acute infections, BVDV infection results in a significant reduction in the white blood cell count in infected animal blood, with the death of immune cells in lymph nodes, which leads to the dysfunction of the animal's immune system, rendering it vulnerable to secondary microbial infections. On the other hand, chronic infection leads to life-long persistent infections (PI) with fetal infection even before the development of the immune system [3, 4].

To avoid the associated economic loss and fetal infection, vaccination has been introduced to control the BVDV infection. Both vaccines, which are modified live virus (MLV) and killed/inactivated vaccines, are well established in the control of the BVDV infection. The MLV vaccine is characterized by a broad long-lasting response; however, it can induce immune dysfunction and intrauterine infection in pregnant animals. On the other hand, the inactivated or killed vaccine showed an incomplete, short-lived immune response and higher safety compared to the MLV vaccine. In addition, the killed vaccine needs adjuvants as the viral antigen is poorly immunogenic. Booster doses of the killed vaccine are required to achieve an effective immune response [5, 6].

Adjuvants play important roles in activating innate immune responses. Ideal adjuvants should induce cell-mediated immune responses and the development of neutralizing antibodies that are specific to the viral antigens with minor or no injection site reactions. The usage of aluminum salts and oil emulsion as veterinary vaccine adjuvants are fit with issues of safety and efficacy [7, 8].

Many adjuvants, such as carbomer, Quil A cholesterol, dimethyldioctadecylammonium chloride commonly known as distearyl dimethyl ammonium chloride (DDA), and montanide oil, were used in combination with the inactivated BVDV vaccine, and these combinations were associated with higher levels of viral neutralizing antibodies and longer-lasting immune responses [9].

Monolaurin is a liposomal natural immune stimulant formed from the esterification reaction between luric acid and glycerol. Many studies tested its antibacterial efficacy besides its role as an adjuvant in various vaccines [10]. Monolaurin is widely used in food and pharmaceutical industries [11]. It can be used as a food additive and dietary supplement due to its immunomodulatory effect. In addition, it can inhibit a wide range of pathogenic bacteria, fungi, and viruses. Monolaurin exerts its anti-inflammatory effect by inhibiting the production of pro-inflammatory cytokines and controlling T-cell proliferation [12–15].

This study aimed to evaluate the efficacy of monolaurin as a novel adjuvant through the preparation of inactivated BVDV vaccines adjuvanted with different concentrations of monolaurin and compare them with the available registered locally-prepared polyvalent vaccine (Pneumo-4) containing the BVDV and adjuvanted with aluminum hydroxide gel.

## Materials and methods

### 1—Viruses and cells

A local Egyptian strain of the bovine viral diarrhea virus genotype 1 (BVD-1, NADL strain, $10^{6.5}$ TICD$_{50}$) was supplied by the Rinderpest-like Diseases Research Department, Veterinary Serum and Vaccine Research Institute (VSVRI), Abbasia, Cairo. It was propagated and titrated on Madin Darby Bovine Kidney cell (MDBK) [16].

## 2—Ethical approval

All procedures involving animals were in accordance with the ethical standards of and approved by the Medical Research Ethics Committee (MREC) of the National Research Centre, Cairo, Egypt.

## 3—Animals

**3–1 White Swiss mice.** Ten Albino Swiss weaned mice weighing 10–15 g were used in running safety tests of the prepared vaccine formulae [17]. They were obtained, reared, and observed in the Laboratory Animals Department, Veterinary Serum and Vaccine Research Institute, Abbasia, Cairo, Egypt. The mice were housed in a ventilated chamber with a 12-hour light-dark cycle (22 ± 1˚C, 60% ± 10% relative humidity). Every day, all groups were observed for any abnormalities or deaths.

**3–2 Calves.** Susceptible healthy native breed male calves aged 6 months were used to evaluate the potency and the safety of different vaccine formulas [17]. Animals were tested using the neutralization test for both BVDV antigens and antibodies to be proof its freedom from both. Animals were isolated in an isolation unit dedicated to the experimental evaluation of the produced vaccines. This isolation unit is located at the Veterinary Serum and Vaccine Research Institute (VSVRI). The unit was well maintained to protect animals from any harm. Reliable and adequate electricity and adequate potable water were available on the premises. The temperature was sufficiently regulated at the room temperature. The unit was adequately ventilated with ample and uniformly-distributed light. Food was supplied *ad libitum*.

## 4—Preparation of monolaurin

The enzymatic production of glycerin monolaurate (GML) was done via the lipase-catalyzed esterification of lauric acid and glycerin in a solvent-free medium according to a previously described technique [18]. The enzymatic method used in the synthesis process had several advantages over the conventional chemical method, including reduced energy and water requirements (S1 File). In addition, the enzymatic esterification reaction is selective and resulted in higher monoglyceride yields [19, 20].

## 5—Vaccine preparation

Briefly, confluent monolayers of MDBK cells grown in Roux bottles were inoculated with the BVDV at a multiplicity of infection (virus to cell, 2:1) and incubated at 37˚C in the presence of 1 mg/mL of trypsin. After 70%–80% of the infected cells showed cytopathic effects, the culture fluid was harvested, clarified, and titrated. The virus was inactivated by stirring with 1% ascorbic acid for 24 hours [21].

Equal volumes of the inactivated virus fluids were thoroughly mixed with 0.5%, 1%, and 2% monolaurin in a 1:1 (vol/vol) ratio, and the pH was adjusted to 7.5. Thiomersal was added as a preservative at a final concentration of 0.001%. Finally, the prepared formulae were distributed in 50-ml sterile bottles and kept at 4˚C until they were used.

## 6—Quality control of the prepared vaccine

**6–1 Sterility test.** A sterility test was performed to ensure that the prepared vaccine is free from microorganisms such as bacteria, fungi, mycoplasma, and viruses, including non-cytopathic BVDVs, according to a previously described technique [22].

**6–2 Safety tests.** Ten Albino Swiss mice were used, five of which were inoculated intraperitoneally with 0.2 ml/mouse of the prepared vaccine, while the other five mice that served

as controls were inoculated with physiological saline. All mice were kept under observation for two weeks to detect any clinical abnormalities [23].

Six male calves, divided into two groups of three each, were used as follows: The first group was inoculated intramuscularly (IM) with ten times the vaccinal dose of the prepared vaccine according to a previously described technique [22]. The other three calves were inoculated with the same dose and inoculated with physiological saline. All animals were kept under observation for two weeks post-inoculation to detect any clinical abnormalities.

**6–3 Potency test.** It was performed according to a previously described technique [22] where 15 calves were used. The animals were divided into five equal groups three each. The first group was injected with the Pneumo-4 vaccine (quadrivalent inactivated vaccine against BVDV, BoHV-1, BPI-3, and BRSV, which is adjuvanted with aluminum hydroxide gel) that was locally prepared in VSVRI, Cairo, Egypt, where animals were injected two doses (5 ml intramuscular) two weeks apart. The other three groups were vaccinated with the prepared vaccines adjuvanted by monolaurin at concentrations of 0.5%, 1%, and 2%, where animals were injected two doses (1 ml intramuscularly) two weeks apart. The fifth group, which served as the control group, was not vaccinated.

Serum samples were collected from all calves on the first day of vaccination (day 0), the second week, the fourth week, and every month up to six months post-vaccination. All serum samples were inactivated at 56˚C for 30 minutes and stored at −20˚C until they were used for serum neutralizing antibody testing.

## 7—Evaluation of neutralizing antibodies

**7–1 Neutralization assay.** All serum samples were tested using the serum neutralization test to detect specific neutralizing antibodies against all vaccinal viruses of the Pneumo-4 vaccine. The assay was applied in micro-titration plates according to a previously described technique [24]. The Serum Neutralizing Antibody titers of the tested serum samples were expressed as log10 values of the reciprocal serum dilution that protects >50% of microtitration plate dilution wells, and these follow the calculation formula of Reed and Muench (1938) [25].

**7–2 Indirect enzyme-linked immunosorbent assay (ELISA).** Seroconversion of vaccinated calves and mice against viral components of the prepared vaccine were estimated via indirect in-house ELISA, with modifications [26]. The results were recorded using a computer-assisted micro plate reader (ELx808TM Absorbance Micro Plate Reader; BioTek Instruments Inc., Winooski, VT, USA).

Checkerboard titrations were performed with 100μl volumes of the antigen, serum, and conjugate to determine optimal dilutions. PBS containing 0.05% Tween 20 was used for washing and dilution. Briefly, BVD antigen preparation was carried out by inoculating the BVD virus on MDBK cells then incubating them at 37˚C for one hour to ensure adsorption, after which the maintenance medium was added. When the cytopathic effect (CPE) reached 70%–80%, freezing and thawing were performed three times to collect the virus, which was kept at −80˚C. The coated plates were blocked by adding 100 μl of blocking buffer (5% skimmed milk) per well and incubating for one hour at 37˚C, after which the plates were washed and dried. Serum samples were added and incubated for one hour at 37˚C and then rinsed and dried. One hundred microliters of peroxidase-labeled rabbit anti-bovine IgG conjugate diluted 1/20000 in PBS Tween was added to each well, incubated for one hour at 37˚C, and then washed. O-phenylenediamine (OPD) was added as a substrate and incubated in the dark for 15 minutes. The reaction was then stopped by adding 25 μl of 1.25 M of sulfuric acid per well and the plates were read using an ELISA reader.

## 8—Statistical analysis

Statistical analysis was done using Microsoft Excel 2010 (Microsoft Corp., Redmond, WA, USA). Quantitative data were expressed as means ± standard deviations. Data were compared between groups of the same virus using the one-way analysis of variance (ANOVA) test. A p-value of ≤0.05 was considered statistically significant.

## Results

The prepared vaccines were found to be sterile, free from any contaminant, and safe for all calves. We found no clinical abnormalities during the experiment.

In the case of the Pneumo-4 vaccine and the vaccine adjuvanted with 0.5% monolaurin, the mean serum neutralizing antibody titers reached the protective levels (NI 0.91, 1.18 & EI 0.91, 1.04, respectively) at the first month post-vaccination and maintained these levels till the fifth month post-vaccination. Regarding the vaccines adjuvanted with 1% and 2% monolaurin, they reached the highest neutralizing antibody titers (NI 0.92, 1.05 & EI 0.94, 1.1) till the end of the experiment, which was the ninth month post-vaccination (Table 1) (Figs 1 and 2) (S1 and S2 Tables). The minimum acceptable titer of the protective level against the BVD virus expressed in log10: 0.9 [27].

## Discussion

The bovine viral diarrhea virus (BVDV), which affects dairy- and beef-producing animals and causes economic losses, is widely distributed in many countries. Vaccination was found to control infection in the case of the absence of persistently infected (PI) animals. The PI animals can shed large amounts of the virus in a herd, which can limit the efficacy of vaccines [28–30]. So, vaccination can result in adequate immunization levels in a herd via the determination of each animal's BVDV status, the implementation of sanitary measures, and the identification and elimination of PI animals [31, 32].

Ideal adjuvants should strengthen the specific immune response against viral antigens with minimal or no immune irritation response [1, 33].

In this study, none of the vaccinated animals developed systemic or local injection-site reactions. According to previous findings, monolaurin was used as an adjuvant with the BVDV

**Table 1. Mean serum neutralizing antibody titers against the BVD virus expressed in NI (neutralizing antibody index) and EI (ELISA index) in vaccinated animals with Pneumo-4 and different vaccine formulas.**

| Time post vaccination | Pneumo-4 | | 0.5% monolaurin | | 1% monolaurin | | 2% monolaurin | | Control unvaccinated | |
|---|---|---|---|---|---|---|---|---|---|---|
| | NI | EI | NI | EI | NI | EI | NI | EI | NI | EI |
| 0 day | 0.02 | 0.18 | 0.0 | 0.05 | 0.0 | 0.01 | 0.01 | 0.01 | Respond negatively lower than protection (0.9) | 0.0 |
| 1 M | 1.72 | 1.84 | 1.39 | 1.6 | 2.06 | 2.19 | 2 | 2.3 | | |
| 2 M | 1.9 | 2.01 | 1.5 | 1.8 | 2.2 | 2.3 | 2.3 | 2.3 | | |
| 3 M | 1.5 | 1.6 | 1.2 | 1.4 | 1.9 | 2.02 | 1.93 | 2.08 | | |
| 4 M | 1.3 | 1.42 | 1.0 | 1.19 | 1.72 | 1.82 | 1.72 | 1.7 | | |
| 5 M | 0.91 | 1.18 | 0.91 | 1.04 | 1.58 | 1.7 | 1.51 | 1.71 | | |
| 6 M | 0.71 | 0.84 | 0.78 | 0.9 | 1.35 | 1.48 | 1.2 | 1.34 | | |
| 7 M | 0.6 | 0.8 | 0.6 | 0.8 | 1.2 | 1.31 | 1.19 | 1.31 | | |
| 8 M | 0.4 | 0.52 | 0.4 | 0.52 | 1.0 | 1.19 | 1.07 | 1.21 | | |
| 9 M | 0.34 | 0.46 | 0.34 | 0.46 | 0.92 | 1.05 | 0.94 | 1.1 | | |

M: month; NI: neutralizing antibody index; EI: Elisa index.

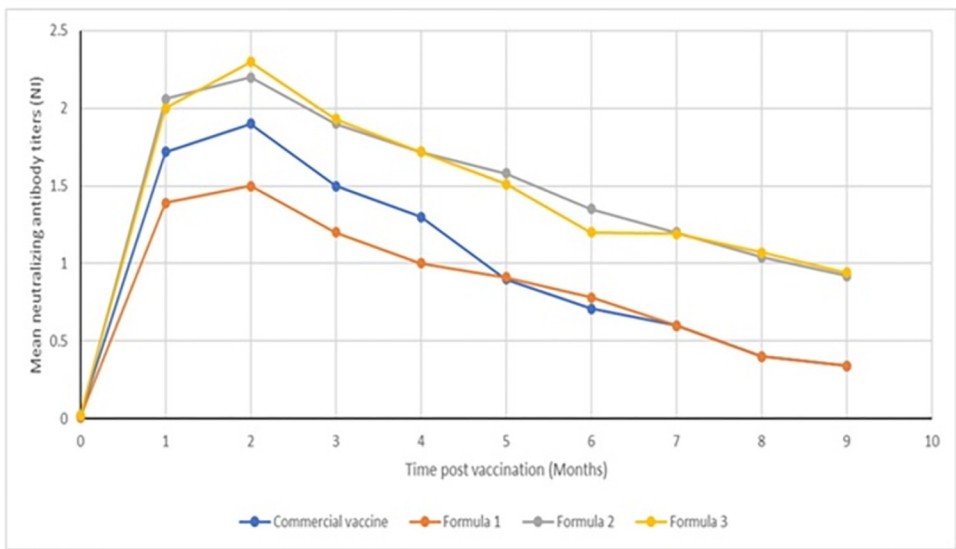

**Fig 1. Neutralizing antibody titers of different BVD vaccine formulas adjuvanted by monolaurin in comparison to Pneumo 4 vaccine.**

vaccine. According to the results of our study, the prepared monolaurin adjuvanted vaccine formulas were sterile and were associated with no systemic or local reactions in the tested animals. We also found that the commercial vaccine adjuvanted by aluminum hydroxide gel produced neutralizing antibodies till the end of the fifth month post-vaccination. Also, the BVDV vaccine adjuvanted by 0.5% monolaurin produced neutralizing antibodies at protective levels till the fifth month, similar to the commercial vaccine. Neutralizing antibody titers peaked after two months post-vaccination. Our results showed that the vaccine adjuvanted with 1% and 2% monolaurin produced protective levels of neutralizing antibodies till the ninth month

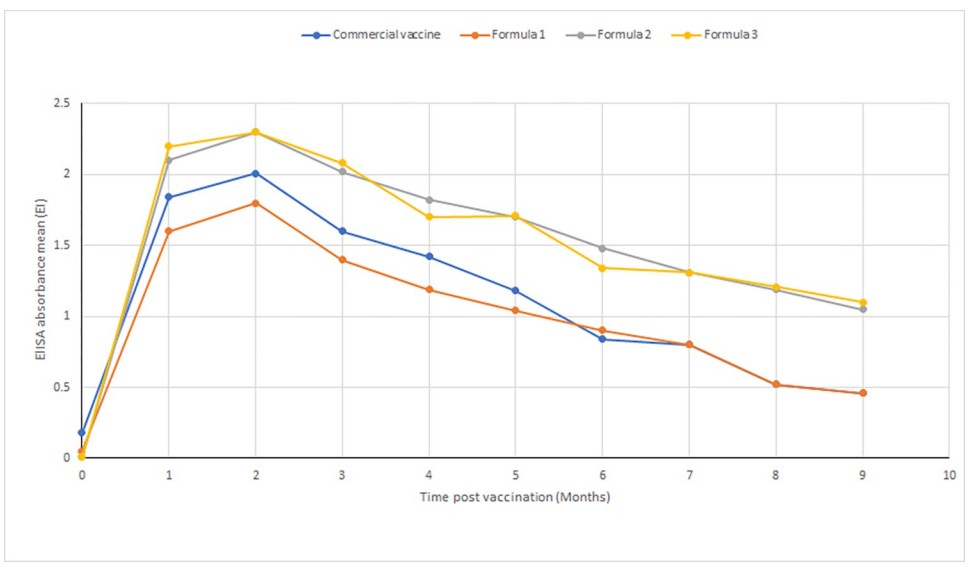

**Fig 2. ELISA results obtained by different BVD vaccine formulas adjuvanted by monolaurin in comparison to the Pneumo-4 vaccine.**

post-vaccination. There were no significant differences between the results obtained by vaccines adjuvanted with 1% and 2% monolaurin. The titers of neutralizing antibodies produced by vaccines adjuvanted with 1% and 2% monolaurin peaked after two months post-vaccination. Many studies have tried different adjuvants to increase the associated vaccine immune response and duration. Ridpath et al. [1] used Quil A cholesterol and dimethyldioctadecylammonium (DDA) bromide as adjuvants with the BVDV vaccine. They found that the prepared vaccines were associated with high neutralizing antibody titers and reduced injection-site inflammation. When montanide oil (ISA 206) was used [16, 34] as an adjuvant, they found that it had a large safety margin with higher neutralizing antibody titers for longer durations compared to the commercial vaccine since it can trap the antigen and release it over long periods and increase the physical presentation of the antigen. The Pneumo-4 vaccine adjuvanted with carbomer 0.5% [17] showed high potency with the stimulation of cellular immunity compared to the vaccine adjuvanted with aluminum hydroxide gel.

## Conclusion

The bovine viral diarrhea virus inactivated vaccine with 1% and 2% monolaurin elicited higher neutralizing antibody titers with longer-lasting effects (nine months) compared to the commercial vaccine adjuvanted by aluminum hydroxide gel. We plan to examine the new adjuvant with the polyvalent vaccine in the near future.

## Supporting information

**S1 File. Preparation of monolaurin.**
(DOCX)

**S1 Table. Neutralizing antibody titers produced by different BVD vaccine formulas adjuvanted by monolaurin in comparison to the Pneumo-4 vaccine.**
(XLSX)

**S2 Table. ELISA results obtained by different BVD vaccine formulas adjuvanted by monolaurin in comparison to the Pneumo-4 vaccine.**
(XLSX)

## Author Contributions

**Conceptualization:** Maha Raafat Abd El Fadeel, Eman M. Soliman, Ahmad Mohammad Allam, Mohamed F. ElKersh, Rehab Mahmoud Abd El-Baky, Ahmad Mustafa.

**Formal analysis:** Eman M. Soliman.

**Methodology:** Maha Raafat Abd El Fadeel, Eman M. Soliman, Ahmad Mustafa.

**Resources:** Rehab Mahmoud Abd El-Baky, Ahmad Mustafa.

**Supervision:** Mohamed F. ElKersh, Rehab Mahmoud Abd El-Baky.

**Validation:** Ahmad Mohammad Allam, Ahmad Mustafa.

**Writing – original draft:** Eman M. Soliman, Mohamed F. ElKersh, Ahmad Mustafa.

**Writing – review & editing:** Ahmad Mohammad Allam, Rehab Mahmoud Abd El-Baky, Ahmad Mustafa.

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
