## [Decision Letter · Decision Letter 0]

24 Feb 2022

PONE-D-21-37629Efficacy and durability of Bovine Virus Diarrhea (BVD) virus killed vaccine adjuvanted with monolaurin.PLOS ONE

Dear Dr. Allam,

Thank you for submitting your manuscript to PLOS ONE. After careful consideration, we feel that it has merit but does not fully meet PLOS ONE’s publication criteria as it currently stands. Therefore, we invite you to submit a revised version of the manuscript that addresses the points raised by reviewer #2 during the review process. His comments are fundamental for a better understanding of the advances and contribution to the this area.

We look forward to receiving your revised manuscript.

Kind regards,

Paulo Lee Ho, Ph.D.

Academic Editor

PLOS ONE

Journal Requirements:

2. In your Methods, please include full details of animal care and housing.

Reviewers' comments:

Reviewer's Responses to Questions

**Comments to the Author**

1. Is the manuscript technically sound, and do the data support the conclusions?

Reviewer #1: Yes

Reviewer #2: No

2. Has the statistical analysis been performed appropriately and rigorously? 

Reviewer #1: Yes

Reviewer #2: I Don't Know

3. Have the authors made all data underlying the findings in their manuscript fully available?

Reviewer #1: Yes

Reviewer #2: Yes

4. Is the manuscript presented in an intelligible fashion and written in standard English?

Reviewer #1: Yes

Reviewer #2: No

5. Review Comments to the Author

Reviewer #1: This is a greatly improved manuscript. The authors have made the necessary corrections throughout the manuscript. The English in particular was greatly improved making the manuscript much more readable.

Reviewer #2: 1-The manuscript needs grammatical revision for a proper understanding. Engligh proof reading should be performed.

2-The experiment is not described in a way which allows a full understanding of a few key aspects: is the commercial vaccine used according to the SPC? What about the groups, how animals were assigned to each group and which criteria were used to know if the groups were equivalent?

3-Do the other antigens in the commercial vaccine have a role in the immune response of the treated animals? In other words, is it correct to compare a monovalent vaccine with a multivalent one to assess the experimental adjuvant?

4-A commercial ELISA has been used to assess the seroconversion, but very little is known about this kit and in particular its modifications, a detailed description of the modifications and the reasons why this test has been chosen is needed

5-SNT technique is used to assess calves before the experiment, but nothing is mentioned about this SNT method. Moreover, what has been done to exclude animals were PI or had other immunosuppressive conditions?

6-What has been done to exclude BVDV circulation during the study? This could also affect the results

6. PLOS authors have the option to publish the peer review history of their article (what does this mean?). If published, this will include your full peer review and any attached files.

Reviewer #1: No

Reviewer #2: No

---

## [Author Response · Author response to Decision Letter 0]

10 Apr 2022

Thank you for your effort in revising the manuscript to meet the journal requirements. Hope the corrections meet your expectation and see the manuscript published soon.

---

## [Decision Letter · Decision Letter 1]

13 May 2022

Efficacy and durability of bovine virus diarrhea (BVD) virus killed vaccine adjuvanted with monolaurin

PONE-D-21-37629R1

Dear Dr. Allam,

We’re pleased to inform you that your manuscript has been judged scientifically suitable for publication and will be formally accepted for publication once it meets all outstanding technical requirements.

Kind regards,

Paulo Lee Ho, Ph.D.

Academic Editor

PLOS ONE

Additional Editor Comments (optional):

Reviewers' comments:

Reviewer's Responses to Questions

**Comments to the Author**

1. If the authors have adequately addressed your comments raised in a previous round of review and you feel that this manuscript is now acceptable for publication, you may indicate that here to bypass the “Comments to the Author” section, enter your conflict of interest statement in the “Confidential to Editor” section, and submit your "Accept" recommendation.

Reviewer #2: All comments have been addressed

2. Is the manuscript technically sound, and do the data support the conclusions?

Reviewer #2: Partly

3. Has the statistical analysis been performed appropriately and rigorously? 

Reviewer #2: Yes

4. Have the authors made all data underlying the findings in their manuscript fully available?

Reviewer #2: Yes

5. Is the manuscript presented in an intelligible fashion and written in standard English?

Reviewer #2: Yes

6. Review Comments to the Author

Reviewer #2: Thank you for the efforts in adressing the comments and questions raised. The manuscript has significantly improved in my view.

7. PLOS authors have the option to publish the peer review history of their article (what does this mean?). If published, this will include your full peer review and any attached files.

Reviewer #2: No

---

## [Editor Report · Acceptance letter]

24 May 2022

PONE-D-21-37629R1 

Efficacy and durability of bovine virus diarrhea (BVD) virus killed vaccine adjuvanted with monolaurin 

Dear Dr. Allam:

I'm pleased to inform you that your manuscript has been deemed suitable for publication in PLOS ONE. Congratulations! Your manuscript is now with our production department. 

Kind regards, 

on behalf of

Dr. Paulo Lee Ho 

Academic Editor

PLOS ONE